# Understanding the Double-Level Influence of Guanxi on Construction Innovation in China: The Mediating Role of Interpersonal Knowledge Sharing and the Cross-Level Moderating Role of Inter-Organizational Relationships

**Yingmiao Qian [1,2], Mengjun Wang [1,*], Yang Zou [2] , Ruoyu Jin [3] , Ruijia Yuan [1,2] and Qinge Wang [1]**

1   School of Civil Engineering, Central South University, Changsha 410075, China;
    154801047@csu.edu.cn (Y.Q.); rj.yuan@csu.edu.cn (R.Y.); wqecsu@126.com (Q.W.)
2   Department of Civil and Environmental Engineering, The University of Auckland,
    Auckland 1010, New Zealand; yang.zou@auckland.ac.nz
3   Department of Built Environment, University of Brighton, Brighton BN2 4AT, UK; R.Jin@brighton.ac.uk
*   Correspondence: Wmjcs@163.com; Tel.: +86-135-0731-5392

**Abstract:** Guanxi, a Chinese term that defines social networks of power and benefits, can be divided into inter-personal and inter-organizational relationships. Guanxi significantly influences construction innovation in China. Many studies have examined the relationship between guanxi and construction innovation at the project or organizational level. However, few of these studies explain how guanxi might affect an individual's innovative behaviour from a double-level perspective. This paper builds on social capital theory and social exchange theory to examine guanxi's role in motivating innovative behaviour in a China-specific construction context. It investigates the main effects of inter-personal relationships on innovative behaviour, the mediating effects of knowledge sharing, and the cross-level moderating effects of inter-organizational relationships. These elements were tested using a survey that received 178 responses from 35 different organizations. The results were analysed using Hierarchical Linear Modelling (HLM) and revealed that inter-personal relationships have positive influences on innovative behaviour, thus highlighting the partial mediating effects of knowledge sharing. In addition, the analyses showed that inter-organizational relationships augment inter-personal relationships and knowledge sharing on innovative behaviour by cross-level interaction. The research findings enhance an understanding of guanxi and innovative behaviour in China-specific construction project settings, as well as verifying the significance of guanxi in stimulating innovative behaviour.

**Keywords:** inter-personal relationships; construction innovation; knowledge sharing; inter-organizational relationships

## 1. Introduction

Today, construction enterprises face intense and increasing competition both globally and regionally [1]. To achieve long-term success, they need to have better productivity and quality control, and leaner production through, among others, technological innovation, operating procedures, organization systems, and procurement [2]. Innovation by definition refers to a significant improvement in a product or service, processes, marketing and organizations [3,4]. The patterns of innovation in construction differ from those within the manufacturing and service activities because

construction activities are context-sensitive and temporary [5]. Previous research has discussed a number of relevant issues, including the models of construction innovation [6,7], the logic and process of innovation in construction [8–10], strategies and public policy of construction innovation [11,12], the ways to implement innovation and the fact that the drivers of innovation are highly related to industry-specific features [13]. Barriers to construction innovation, such as temporary project-based organization, lack of knowledge sharing, the conservation of established practices, perceived high financial investment needed in innovation, and limited resources [14], have led to the view that the construction industry is conservative and less innovative [8]. To address this view, prior studies have focused on antecedent variables that affect innovation at the project-based organizational level [15] and the individual level [5], and the construction innovation process at the project level [10]. Nevertheless, research on construction innovation at diverse levels remains in its infancy. Studies on construction innovation in China at the individual/organization level are very limited. Considering the peculiarities of the China-specific construction innovation context (e.g., renqing (emotional) society, recent deregulation, and a booming construction industry), there is a need to understand the influence of guanxi on construction innovation at the inter-personal and inter-organizational levels.

Guanxi arises from Confucian ideology and refers to the notion of a relation-centred and collaborative culture that seeks relationship harmony. As such, guanxi has profound implications for business transactions amongst Chinese communities [16]. For this reason, both academics and industrial practitioners have dedicated much attention on the influence of guanxi on the individuals' ability and level of cooperation [17–20], and this has gradually extended to investigating the relationship between guanxi and innovation. For example, Chu et al. (2018) pointed out that external relationships are important suppliers of resources and knowledge in logistics service innovation, and suggested that both political and business guanxi have a positive effect on logistics service innovation [21]. Meanwhile, guanxi involves the exchange of social obligations and the asking for and provision of favours [18]. It helps a firm acquire scarce resources, business information and opportunities, and enhances the firm's advantage in terms of performance and innovation [22]. While extant research on guanxi has extensively examined the effects of relationships on innovation at a single firm's level [23], the influences on individual innovation behaviour and interpersonal relationships have been overlooked, resulting in a research gap in the construction innovation literature. To address these limitations, this study focuses on antecedents (i.e., inter-personal relationships, knowledge sharing and inter-organizational collaborative relationships) with an individual's innovative behavior as the output variable. It also examines how guanxi influences innovation in construction.

The main objective of this study is to acquire an intensive understanding of the influence of guanxi on individual innovative behaviour in construction projects, and to reveal the nature of the mediating role of knowledge sharing and the cross-level moderation role in inter-organizational collaborative relationships. The research questions are how guanxi influences construction innovation at double levels, and what is the role of knowledge sharing in construction innovation. The theoretical and practical contributions of this study include: (i) inter-personal relationships act as the precursor to knowledge sharing and innovative behaviour, while knowledge sharing partially transmits the influence of inter-personal relationships on innovative behaviour; and (ii) inter-organizational collaborative relationships act as the moderation mechanism, whereby the cross-level influence of inter-personal relationships on innovative behaviour through knowledge sharing is enhanced.

To sum up, most social behaviors and institutions in China are deeply influenced by social guanxi and can be analyzed through social guanxi [20]. In addition, construction innovation, and those individuals and organizations (owners, designers, constructors, material suppliers, equipment manufacturers, consulting agencies) involved in construction innovation are embedded in different social guanxi, and their decisions and behaviors are deeply affected by guanxi. Therefore, this study establishes a concept model to introduce guanxi into construction innovation management, and by employing a contingent model, it tests how the interaction of the two groups (inter-personal relationships and inter-organizational relationships; knowledge sharing and inter-organizational

relationships) to influence individual innovative behaviour. This study provides a more integrative view of how to stimulate individual innovation in construction projects by facilitating knowledge sharing and improving relationships between team members and stakeholders.

## 2. Theoretical Background, Research Hypotheses and Conceptual Model Development

### 2.1. Theoretical Background

In Joseph Schumpeter's opinion, innovation is viewed as determining new combinations and setting up new production functions [24,25]. This theory of innovation has attracted much attention from scholars and institutions, and has contributed to refining the definition of innovation. For instance, Damanpour (1992) defined innovation as the adoption of a new idea or behaviour [26], and the Department of Trade and Industry in the UK (2007) described innovation as the successful exploitation of new ideas [27]. The context-sensitive nature of construction and the variety of organizations involved in construction means that the patterns of construction innovation are different from those in the manufacturing sector and in services [5]. Dikmen et al. (2005) defined construction innovation as a system in which the elements are objectives, strategies, environmental barriers/drivers, and organizational factors [28]. Because of the increasing complexity and uncertainty of construction innovation, it is necessary to modify the paradigm so that it includes collaborative innovation in order to understand and implement it in a China-specific context. Construction innovation in China is known to be collaborative in nature, i.e., the organizations in construction seek reciprocal collaboration at various stages of innovation [29,30], which can be across organizational boundaries through the sharing of knowledge, ideas and expertise [31,32].

Social capital refers to all the resources embedded in social network relationships [33], which implies that social actors engaging in such relationships can obtain access to resources to further their own interests [34]. The social capital theory emphasizes the exchange of non-financial resources, establishment of common resources [35], and that the exchange partners have a responsibility to mutually contribute valuable resources that may be helpful [36]. Thus, by utilizing social capital, actors (e.g., individuals, organizations, and commercial entities) can gain indispensable external resources that promote innovation and enhance performance. Guanxi, a China-specific concept that dominates business activities throughout the country [16], has been closely related to the western culture concept of social capital; consequently, guanxi has attracted the attention of scholars in management and business fields. Some of them have found that guanxi produces significant effects on technological innovation [37], and innovation performance [38].

Social exchange theory postulates that all social behaviours result from an exchange process [39], and an important assumption of the theory is that the behaviours are based on reciprocal exchanges [40, 41]. In essence, social exchange theory is one of the most influential conceptual paradigms applied to understanding workplace behaviour [42], exchange rules and norms that shape social behaviours, and resource exchanges [43]. Furthermore, social exchange tends to generate emotions related to individual obligation, gratitude and trust [39], which may influence personal innovation behaviour. Knowledge sharing, a specific pattern in social exchange, also has an impact on innovation and has been investigated by several scholars [44–46]. Innovation practices in construction projects tend to rely heavily on an individual's knowledge, skill and experience. Meanwhile, knowledge sharing activities, as important ways to improve personal knowledge, can be simultaneously seen as necessary for innovation in the construction process.

### 2.2. Research Hypotheses

#### 2.2.1. Main Effect: Inter-Personal Relationships and Innovative Behaviour

Guanxi is viewed as an intimate and common relationship amongst individuals or organizations via high-quality social activities and reciprocal interest exchanges [47]. Inter-personal relationships

are a complex notion and comprises emotions and feelings toward others [2]. The family tie is a fundamental pattern in inter-personal relationships, and the scope of this tie can be extended to other social groups, such as kin, friends, and acquaintances [48]. Thus, people can develop inter-personal relationships within families, friends, classmates, colleagues and so on. Good interpersonal relationships mean that there is at least a kind of guanxi within families, friends, classmates or colleagues. Meanwhile, inter-personal relationships are also widely recognized as assets at a business level [49], allowing firms to acquire and sustain a competitive advantage. If effectively utilized, inter-personal relationships can cut cross organizational boundaries by providing an alternative, informal and efficient network to get resources. For instance, Chen et al., (2015) affirmed that Chinese entrepreneurs could gain information and resources via their guanxi networks, thereby influencing a firm's success [50].

Furthermore, many scholars have stressed that inter-personal relationships are a key variable for innovation. During an analysis of a firm's innovation, Arribas et al., (2013) pointed out that guanxi, as a type of social capital, can have a deep influence on innovation and performance [51]. Wang and Chen (2018) found that if there are close inter-personal relationships, individuals are more willing to support and encourage innovative ideas because familiarity provides the confidence that assists in changing ideas into innovative outcomes [52]. Holmen et al., (2005) recognized inter-personal relationships among partners as an informal guarantee that can have a positive influence on innovation [53]. To sum up, based on collaborative efforts in construction innovation, inter-personal relationships can promote more intense interactions among partner firms' personnel, allowing them to be more willing to create and share new ideas, thereby enhancing personal innovation behaviour. This study thus proposes the existence of a positive relationship between inter-personal relationships and innovative behaviour in construction project settings.

**Hypothesis (H1).** *Inter-personal relationships have a positive influence on innovative behaviour in construction projects.*

2.2.2. Mediating Effect: Knowledge Sharing

Inter-personal relationships and knowledge sharing

It is accepted that knowledge sharing is an activity applicable at the individual, group, and organizational level [54,55]. In the present study, knowledge sharing refers to individuals' knowledge exchange activities and focuses on the process of knowledge acquisition, exchange, and diffusion [44] amongst individuals from diverse organizations involved in a construction project, which, in turn, contributes to knowledge creation and construction innovation. Consequently, knowledge sharing can be seen as a non-institutional arrangement that may not be motivated by direct economic incentive rewards [55], but more easily inspired by individual self-satisfaction and harmony with others. Moreover, inter-personal relationships will play a vital role in knowledge sharing due to the latter being non-spontaneous. In a discussion pertaining to Taiwan's high-tech industry, Wang et al., (2012) revealed that inter-personal relationships could have a positive influence on knowledge sharing, emphasizing that high-quality inter-personal relationships shape employees' intentions to share and exchange knowledge [56]. Similarly, Cao and Xiang (2012) claimed that guanxi served as a mediator between knowledge governance and knowledge sharing, suggesting that firms need to foster a harmonious atmosphere in order to enhance the positive influences of inter-personal relationships [57]. Therefore, employees who have high-quality guanxi with colleagues in construction innovation will tend to share their knowledge and experience as a way of demonstrating this mutually supportive relationship. On the basic of these previous findings, this study postulates that knowledge sharing is positively related to inter-personal relationships in the process of construction innovation.

**Hypothesis (H2).** *Inter-personal relationships have a positive influence on knowledge sharing in construction innovation.*

Knowledge sharing and innovative behaviour

Given that innovation in construction is fundamentally a collaborative practice [52], individual innovative behaviour embodied in a complex construction project context demands the contribution of knowledge from diverse professional technicians. From this perspective, knowledge sharing is an efficient way to implement innovation in construction, and it is obvious that the capability of individuals to exploit and absorb knowledge may determine the level of innovation [44]. According to social exchange theory, knowledge sharing can be viewed as a social exchange behaviour [58], involving collaborative knowledge exchange between diverse individuals in order to solve new problems, improve decision-making processes and achieve innovation [59,60]. Overall, it is significant that employees, to facilitate their innovative activities, may be willing to share knowledge externally as well as internally within an organization [45].

Accordingly, many scholars have shown intense interest in the link between knowledge sharing and innovation. For example, Abou-Zeid and Cheng (2004) pointed out that two perspectives of innovations (thing-oriented and process-oriented) are positively related to knowledge management, especially to knowledge exchange [61]. Swan (2007) analyzed how knowledge management could promote innovation from diverse viewpoints: production, process and practice [62]. Furthermore, in relation to supply chain networks, Wang and Hu (2017) claimed that knowledge sharing serves as a partial mediator between innovation activities and innovation performance, and stated that firms that share knowledge are more likely to engage in more inter-firm collaborative innovations that generate higher levels of performance [63]. In previous studies on the relationship between knowledge sharing and innovation, the authors concentrated their attention primarily at the firm level and supply chain network [62,63], so studies that have focused on construction innovation are relatively rare. To fill the gaps in the current research, this study proposes the following hypothesis:

**Hypothesis (H3).** *Knowledge sharing has a positive influence on innovative behaviour in construction projects.*

Moreover, if H1 and H2 are tenable, then knowledge sharing will act as a mediator between inter-personal relationships and innovative behaviour. Consequently, a fourth hypothesis is proposed:

**Hypothesis (H4).** *Knowledge sharing has a mediating role in the effect of inter-personal relationships on innovative behaviour.*

2.2.3. Cross-Level Moderating Effect: Inter-Organizational Relationships

Inter-organizational relationships (IOR), established by frequent interactions between two or more organizations [64], are generally seen as enduring transactions and connections that occur among these organizations [65,66]. From a resource-based perspective, IOR are able to assist organizations, in their quest for competitive advantage, and to obtain mutual benefits via reciprocating resources they could not acquire by themselves [67,68]. From transaction cost theory, IOR tend to decrease transaction costs by providing an informal and effective network systems that can help sustain organizational interests [69]. Currently, there are two types of IOR, formal and informal, which are increasingly dominant across construction industries. Formal IOR are rooted in contract legalities, and informal IOR are rooted in trust and commitment. Due to opportunism, informal IOR in a construction project are more efficient for innovation than formal IOR.

Partner's commitment to cooperate with one another has been widely regarded as one of the key determinants in establishing long-term relationships amongst diverse organizations [70], reflecting the organizations' intentions to sustain long-term partnerships [70,71]. Inter-organizational commitment can promote the smooth coordination of management practices between different parties [72], especially in innovative activities where inter-organizational commitment could reduce innovative risks. Commitment between organizations is helpful in addressing the free rider problem of innovation that is a frequent phenomenon in the construction industry. Inter-organizational commitment also

tends to create more united construction innovation to cope with innovation tasks, and strives to fulfil innovation goals via the effective integration of individuals' innovative behaviour. Thus, this study puts forward the following hypotheses:

**Hypothesis (H5).** *Inter-organizational commitment serves as a cross-level moderator that can amplify the influence of inter-personal relationships on innovative behaviour.*

Inter-organizational trust is critical in construction innovation [73]. Construction innovation can be developed by employees' collaborative efforts via utilizing and integrating knowledge, experiences and skills. Previous research concluded that the level of inter-firm trust can impact on information communication and knowledge sharing between firms, thus affecting innovation [74]. The greater the trust amongst government agencies, owners, designers, construction units, suppliers of materials and equipment, and research institutions, the greater the willingness to share knowledge for forming new ideas. As a consequence, there is a higher likelihood of accentuated innovative behaviour [5]. In contrast, lower trust leads to less knowledge sharing and reduced innovative behaviour. Therefore, a high level of inter-organizational trust can have a positive influence on individuals' knowledge sharing and innovative behaviour; thus, the following hypothesis is posited.

**Hypothesis (H6).** *Inter-organizational trust exerts a cross-level positive moderating influence on the connection between knowledge sharing and innovative behaviour.*

*2.3. Conceptual Model Developement*

Based on the theoretical background and research hypotheses, the conceptual model of the study is illustrated in Figure 1. Those involved in construction projects will better communicate with each other due to inter-personal relationships and will be more willing to share knowledge, leading to increased innovative behaviour at the individual level. Thus, innovative behaviour will be associated with better inter-personal relationships and knowledge sharing, knowledge sharing will have a mediating role on the effect of inter-personal relationships on innovative behaviour, and inter-organizational relationships will act as cross-level moderators to influence hypothesis 1 and hypothesis 3.

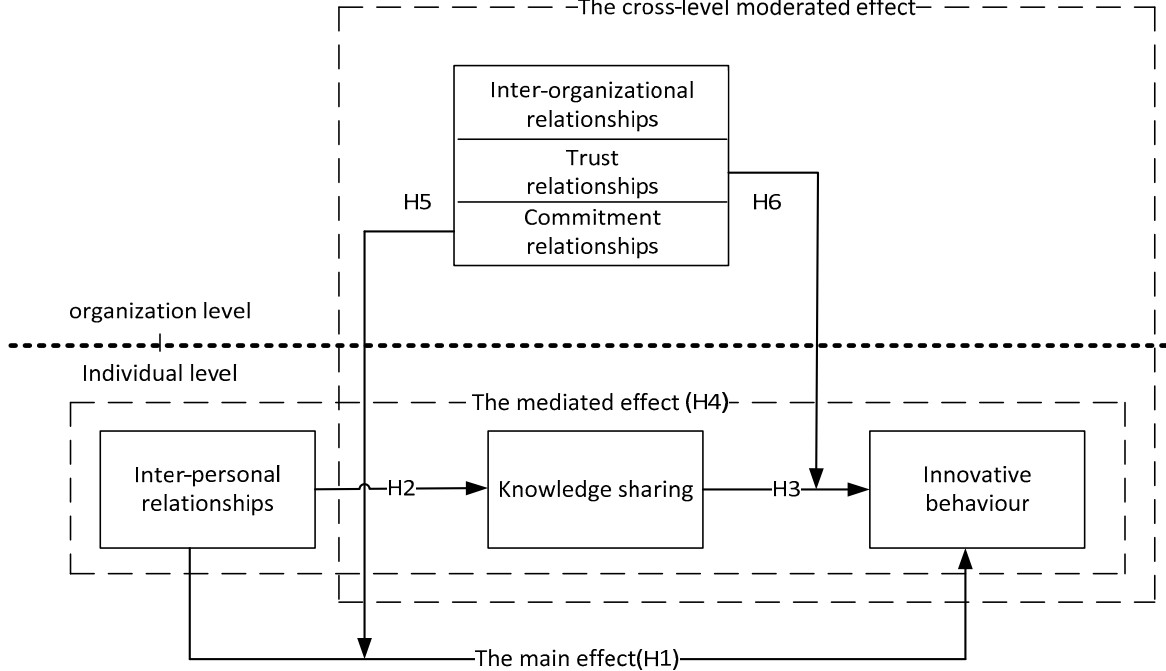

**Figure 1.** The hypothesized conceptual model.

Further, there are some differences and connections between the conceptual model and social capital theory. One difference is that the model and theory are generated in diverse cultural backgrounds: social capital theory originates from the West and the hypothesized conceptual model is unique to China. Chinese society attaches great importance to guanxi, and guanxi can lead to the formation of different social networks, thereby helping to obtain innovation resources (knowledge) and to promote construction innovation. Consequently, the guanxi model is able to be developed in the Chinese construction industry. Besides, guanxi is frequently seen as social capital in China, and emphasizes that social actors engaging in such relationships can obtain access to the resources for their own interests (construction innovation), which is the purpose of utilizing guanxi.

The striking feature of this model is the interaction of guanxi at different levels in the context of Chinese culture, which makes guanxi suitable for the analysis of individuals' innovative behavior in the Chinese construction industry. Construction innovation generally involves many individuals and organizations and guanxi at different levels, such as inter-personal relationships and organization relationships. These enable the development of extensive construction innovation networks and the gathering of heterogeneous innovation resources at different levels, thereby improving individuals' innovation efficiency.

## 3. Research Methodology

### 3.1. Design of Questionnaire

The questionnaire survey is a common and effective way to conduct qualitative research, and has been extensively implemented in innovation research [2,5]. Thus, a questionnaire survey method was utilized in this study to gather professional perspectives on construction innovation management. To obtain the measurement scales of the questionnaire, a wide literature review and interviews were conducted to support the development of the questionnaire survey [75]. Ten Chinese specialists with senior titles and extensive innovative experience were interviewed, via a structured format, to understand the antecedent factors they deemed could influence innovative behaviour in construction projects. The interviews lasted up to 1–2 h per specialist. Several factors such as inter-personal relationships, knowledge sharing and inter-organizational relationships emerged from the analysis of the interview content (See Appendix A for structured interview questions). Then, based on the literature review, details of these several factors (see Appendix B) were obtained. The detailed measurement scale is analyzed in Section 3.3.

The questionnaire, developed from the literature review and initial interviews, was separated into two parts. The first part consisted of respondents' personal information (i.e., gender, education level and working life) and measured the respondent's innovative behaviour (I.I.N., containing five items). The second part measured three antecedents of innovative behaviour, including inter-personal relationships (I.R., containing five items), knowledge sharing (K.S., containing four items) and inter-organizational relationships (I.O.R, containing nine items). This content ensured the questionnaire was appropriate for this research.

### 3.2. Sample Distribution, Questionnaire Release and Recycling

#### 3.2.1. Sample Distribution

Because it is a project-based industry, construction involves many participants, including government agencies, owners, designers, construction units, suppliers of materials and equipment, universities and scientific research institutions, and each has diverse roles in the process of construction innovation [76]. Thus, to ensure the coverage of the questionnaire and to ensure the survey was representative, the respondents came from these key participating groups. In addition, the Hierarchical Linear Model (HLM) is often used to analyze the interaction of variables between different levels, such as the individual level and organizational level. These organization-level samples should contain

at least 30 organizations [77]. As shown in Figure 2, these participants came from 35 organizations and had to fulfill the following conditions: (1) belonged to a basic functional unit in construction innovation; (2) had considerable experience of construction innovation or innovation management; and (3) frequently worked with some of the other participants. After many research seminars with Chinese experts on construction innovation, the selection of these conditions was derived from their understanding and suggestions on construction innovation.

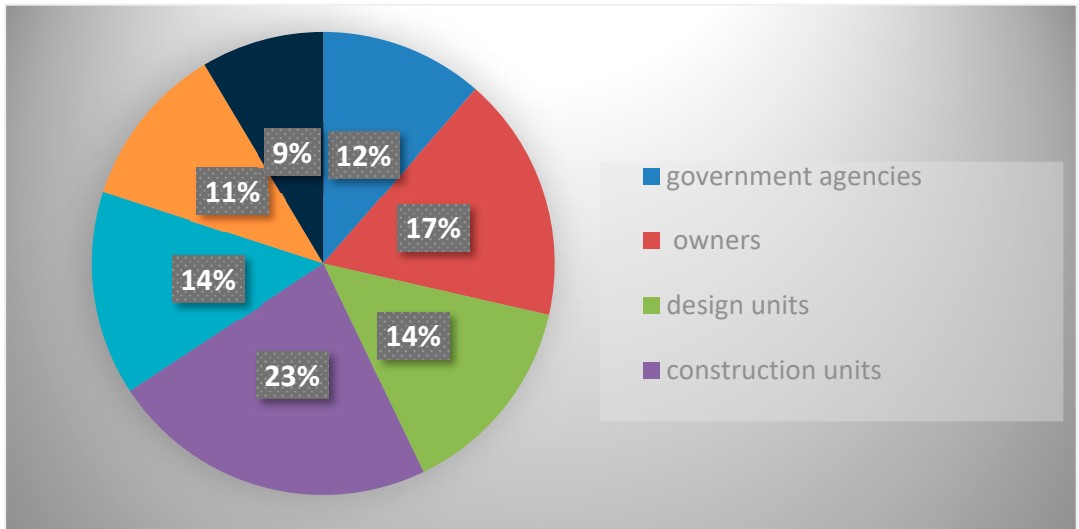

**Figure 2.** The construct of survey respondents.

3.2.2. Questionnaire Release and Recycling

During 2017–2018, the questionnaires were released and recycled in two stages under the guidance of one of the authors. The first stage was to evaluate the questionnaire quality through releasing questionnaires to ten Chinese specialists in construction innovation, thereby allowing for revision of the questionnaire. The second stage was to formally release the questionnaires to the 35 organizations by email, online or in person. The respondents at different levels were asked to consider a range of items; for instance, project managers in diverse organizations focused on the items related to inter-organizational relationships, knowledge sharing and innovative behaviour, while professional technicians focused on the items related to inter-personal relationships, knowledge sharing and innovative behaviour.

The survey respondents were asked to estimate all variables using a 5-point Likert scale, a frequently employed scale that has been applied in previous research [5,78,79]. The data were collected at two levels: from managers representing the organizational level and from employees in different organizations in order to minimize any bias [80,81]. Eventually, 245 questionnaires were disseminated for the study, and after finishing a careful review of the collected questionnaires, the research team found that 178 of the responses could be considered valid. The fundamental information from the respondents is depicted in Table 1.

**Table 1.** The fundamental information from the respondents.

| Items | Gender | | Working Experience(years) | | | | Education Level | | | | |
|---|---|---|---|---|---|---|---|---|---|---|---|
| | Male | Female | Less than 5 | 6–10 | 11–20 | More than 20 | Under Junior College | Junior College | Bachelor | Master | Ph. D and above |
| Numbers | 149 | 29 | 11 | 84 | 49 | 34 | 2 | 23 | 101 | 38 | 13 |
| Percentage | 83.7% | 16.3% | 6.4% | 47.0% | 27.3% | 19.3% | 1.3% | 13.1% | 56.8% | 21.3% | 7.5% |

*3.3. Measurements*

The measurement of all variables in the present study is provided in this section. All the survey questionnaires were translated from Chinese to English because the majority the respondents were Chinese. All the items in the questionnaire were estimated by the respondents on a 5-point Likert scale with anchors from 1 (strongly disagree) to 5 (strongly agree). Measurement items for each construct of the four latent variables are listed in Table A1.

3.3.1. Individuals' Innovation Behaviour (I.I.B)

Construction innovation is considered to be the collaboration of individuals' innovative behaviour in different organizations, and as the main dependent variable, the individuals' innovation behaviour (I.I.B) consisted of four items that were developed from Zhang et al. [5], and Scott and Bruce [82]. The structure of this variable was measured at the individual level via asking professional technicians various questions such as: "The members in project-based organizations always generate creative ideas or new solutions" The responses ranged from 1 to 5 with higher scores suggesting that individuals were more innovative.

3.3.2. Inter-Personal Relationships (I.R)

Inter-personal relationships are seen as intimate and common relationships amongst individuals [47]. In light of this observation, inter-personal relationships were measured using five items validated by Zhang and Hartley [2], and the respondents were asked to express their agreement with statements such as: "My organizational main technicians in a construction project have good personal relationships with other technicians from other organizations in construction innovation". The responses with higher scores indicated that the influences of inter-personal relationships were positive.

3.3.3. Knowledge Sharing (K.S)

Knowledge sharing, as a key factor for effecting innovation, was measured through four items adapted by Cheng and Li [83]. The representative sample statement was "We are willing to share information or ideas with the other members of a project-based organization", and the responses with higher scores indicated that inter-organizational relationships were positive.

3.3.4. Inter-Organizational Relationships (I.O.R)

Inter-organizational relationships generally focus on trust and commitment, so based upon this the trust between organizations was measured via five items validated by Rodríguez et al. [84]. One sample item was "We believe the information that this partner provides us". The commitment between organizations was measured via four items validated by Gu et al. [85]. Another sample item was "We are committed to this partner". The responses with higher scores indicated that inter-organizational relationships were significant.

*3.4. Analytical Procedure of Results*

Analysis of the results obtained from the questionnaire survey was undertaken in three phases as follows.

Firstly, in the preliminary analyses of the data, a reliability test and exploratory factor analysis (EFA) for each measurement item were conducted to estimate whether the structure of the variables was in accord with the anticipated results. The software SPSS22.0 was utilized to carry out the reliability test and EFA, thereby allowing for a discussion of the results for these measurement items.

Secondly, confirmatory factor analysis was conducted to validate the distinctiveness of these variables, including inter-personal relationships, knowledge sharing, inter-organizational relationships and innovative behaviour. The software Amos17.0 was utilized to examine the model's measurement

of the latent variables, and the results suggested that the hypothesized model with four latent variables excellently fitted our data when compared to other models.

Thirdly, given the multi-level characteristics of our data, it is necessary to discriminate the variance at the individual and organizational levels in examining the hypotheses. Thus, hierarchical linear modelling (HLM) was applied to test our research hypotheses, including the main effect as well as the mediating and moderating effects, through using the software HLM version 6.08. Subsequently, innovative behaviour and knowledge sharing were regarded as dependent variables. The main effect was examined with inter-personal relationships, and knowledge sharing was seen as the mediator in the mediating effect testing, and inter-organization relationship was regarded as the moderator in moderating effect testing. All the results were estimated according to the significance of the coefficients and R-square.

The collected data was then analysed by following the designed process (Figure 3) to help assess whether the scales satisfied the requirements of reliability, validity, and to test the hypotheses.

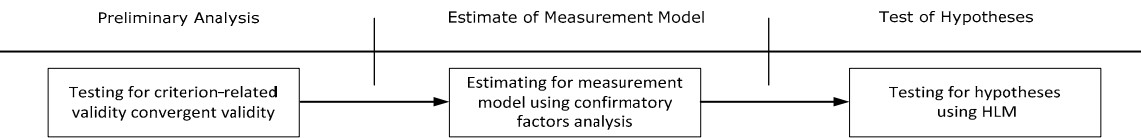

**Figure 3.** The procedure for analyzing the results.

## 4. Results

### 4.1. Preliminary Analysis of Data

The descriptive statistics, intra-class correlation coefficients (ICC), and inter-correlations amongst all the variables at individual and organizational levels are shown in Table 2. Specifically, independent variables at individual levels displayed a statistically positive relationship with innovative behaviour, and the inter-organizational relationship was positively related to innovative behaviour. As anticipated, ICC values for each measure were high, suggesting that there is a significant influence of inter-organizations on individual ratings, and providing the justification for modelling inter-organizational relationship as Level 2 measures.

**Table 2.** Means, standard deviations (SD), and correlations of the study variables.

| Variable | Mean | SD | ICC | 1 | 2 | 3 | 4 | 5 |
|---|---|---|---|---|---|---|---|---|
| Individual level [a] | | | | | | | | |
| 1. Innovative behaviour | 3.16 | 0.59 | 0.27 | 1 | | | | |
| 2. Inter-personal relationships | 3.28 | 0.66 | 0.12 | 0.42 ** | 1 | | | |
| 3. Knowledge sharing | 3.37 | 0.72 | 0.39 | 0.59 ** | 0.47 * | 1 | | |
| Organizational level [b] | | | | | | | | |
| 4. Inter-organizational commitment relationship | 3.64 | 0.47 | 0.17 | 0.40 * | 0.52 ** | 0.47 ** | 1 | |
| 5. Inter-organizational trust relationship | 3.57 | 0.43 | 0.15 | 0.37 * | 0.49 ** | 0.41 ** | 0.31 * | 1 |

Notes: a: n = 178 members, b: n = 35 organizations; *, ** Correlation is significant at the 0.05 and 0.01 levels (two-tailed), respectively.

### 4.2. Measurement Model Estimating

Based on the utilization of Confirmatory Factor Analysis (CFA), the efficiency of the hypothesized model was estimated, and the factor structure of the items was examined. Indices such as the Tucker–Lewis index (TLI), comparative fit index (CFI) and standardized root mean square residual (SRMR) were utilized to estimate the model fit. As shown in Table 3, the CFA results suggested that the model, with its four latent variables including inter-personal relationships, knowledge sharing,

inter-organizational relationship and innovative behaviour (Model IV), demonstrated an excellent fit when compared to alternative models (Models I–III); all other alternative models resulted in a poorer fit, due to having high $\chi^2$/df, SRMR values, and low TLI and CFI values.

**Table 3.** Confirmatory factor analysis (CFA) results of measurement model.

| Model | Description | $\chi^2$/df | SRMR | TLI | CFI |
|---|---|---|---|---|---|
| Model I | One factor: all items loading upon the same single factor, (innovative behaviour with guanxi and knowledge sharing) | 5.891 | 0.097 | 0.586 | 0.593 |
| Model II | Two factors: guanxi (integrated: interpersonal relationships and organizational relationships) and innovative behaviour with knowledge sharing | 5.233 | 0.086 | 0.667 | 0.674 |
| Model III | Three factors: interpersonal level variable (integrated guanxi and knowledge sharing), inter-organizational relationship and innovative behaviour | 3.926 | 0.078 | 0.751 | 0.773 |
| Model IV | Four factors: inter-personal relationships, knowledge sharing, inter-organizational relationship and innovative behaviour | 2.752 | 0.051 | 0.869 | 0.871 |

Notes: N = 178, there were widely acceptable thresholds to evaluate the model fit indices, for instance, nearly 0.90 is a good fit for TLI and CFI and 0.08 is a good fit for SRMR [86].

*4.3. Research Hypotheses Testing*

4.3.1. Steps for Testing the Research Hypotheses

Table 4 shows the results of the research hypotheses testing. The research hypotheses were examined in three steps as follows.

The first step was to examine the fitness of this multilevel analysis, thus, the null model should previously be established. The results showed meaningful inter-organizational variance ($\chi2[35] = 17.2$, $p < 0.001$) for innovative behaviour. Meanwhile, the evaluation of ICC indicated that 15.1% of the variance in innovative behaviour was between level 2 (organizational level) and level 1 (individual level); thus, the multilevel analysis was a fit for the data.

The second step was to examine the main and mediating effects at individual level and this involved four formulas: (1) innovative behaviour = $\beta1 + \beta2 \times$ inter-personal relationships + $\varepsilon1$ (Hypothesis 1, see Model 1 with Y = innovative behaviour as an outcome in Table 4); (2) knowledge sharing = $\beta3 + \beta4 \times$ inter-personal relationships + $\varepsilon2$ (Hypothesis 2, see Model 2 with Y = knowledge sharing as an outcome in Table 4); (3) innovative behaviour = $\beta5 + \beta6 \times$ knowledge sharing + $\varepsilon3$ (Hypothesis 3, see Model 3 with Y= innovative behaviour as an outcome in Table 4); and (4) innovative behaviour = $\beta7 + \beta8 \times$ inter-personal relationships + $\beta9 \times$ knowledge sharing + $\varepsilon3$ (Hypothesis 4, see Model 4 with Y = innovative behaviour as an outcome in Table 4).

The third step was to examine the moderating effects of this study at the cross-level, and the following are the key formulas for level 1 and level 2 models for innovative behaviour: (1) innovative behaviour = $\beta7 + \beta8 \times$ inter-personal relationships + $\beta9 \times$ knowledge sharing + $\varepsilon3$ (at individual level); (2) $\beta7 = \gamma00 + \gamma01 \times$ inter-organizational commitment/trust relationship+u0; (3) $\beta8 = \gamma10 + \gamma11 \times$ inter-organizational commitment relationship+u1; and (4) $\beta9 = \gamma20 + \gamma21 \times$ inter-organizational trust relationship+u2 (at organizational level). While $\beta7$, $\beta8$, $\beta9$ at organizational level was substituted into the individual level, the whole model could be acquired by innovative behaviour = $\gamma00 + \gamma01 \times$ inter-organizational relationship + $\gamma10 \times$ inter-personal relationships + $\gamma11 \times$ inter-personal relationships *inter-organizational commitment relationship + $\gamma20$ knowledge sharing + $\gamma21 \times$ knowledge sharing *inter-organizational trust relationship + $\varepsilon4$.

**Table 4.** The results of research hypotheses testing.

| Model | Coefficient (SE) | | | | | | R2 |
|---|---|---|---|---|---|---|---|
| | Intercept | I.R | K.S | I.O.C.R/I.O.T.R | I.R*I.O.C.R | K.S*I.O.T.R | |
| Model 0 [a] | 3.617 (0.042) ** | | | | | | 0.459 |
| | | H1: The effect of inter-personal relationships on innovative behaviour | | | | | |
| Model 1 [a] | 3.613 (0.043) *** | 0.412(0.037) *** | | | | | 0.513 |
| | | H2: The effect of inter-personal relationships on knowledge sharing | | | | | |
| Model 2 [a] | 3.426(0.039)*** | 0.370(0.051) *** | | | | | 0.509 |
| | | H3: The effect of knowledge sharing on innovative behaviour | | | | | |
| Model 3 [a] | 3.613 (0.043) *** | | 0.473(0.061) *** | | | | 0.672 |
| | | H4: The mediation of inter-personal relationships and innovative behaviour by knowledge sharing | | | | | |
| Model 4 [a] | 3.613 (0.043) *** | 0.156(0.077) * | 0.547(0.062) *** | | | | 0.736 |
| | | H5: Moderator effect of inter-personal relationships and innovative behaviour | | | | | |
| Model 5 [b] | 3.613 (0.043) ** | 0.276(0.056) * | 0.326(0.054) *** | | | | 0.827 |
| Model 6 [b] | 3.613 (0.043) ** | 0.276(0.056) * | 0.326(0.054) *** | 0.296(0.084) + | 0.353(0.073) + | | 0.735 |
| | | H6: Moderator effect of knowledge sharing and innovative behaviour | | | | | |
| Model 7 [b] | 3.613 (0.043) ** | 0.276(0.056) * | 0.326(0.054) *** | 0.2740.091) + | | 0.341(0.085) + | 0.752 |

Notes: N = 178; Standardized beta coefficients and unstandardized intercept value are reported. I.R, inter-personal relationships; K.S, knowledge sharing; I.O.R, inter-organizational relationship; [a], at individual level; [b], at organizational level; *** $p < 0.001$, ** $p < 0.01$, * $p < 0.05$, + $p < 0.1$.

### 4.3.2. Main and Mediating Effects of This Study

The main and mediating effects in this current study are shown by the results registered for Model 1–Model 4 in Table 4. Hypothesis 1 postulated that inter-personal relationships have a positive influence on innovative behaviour in construction projects, which was the main effect of this study. The results indicated that inter-personal relationships significantly related to innovative behaviour ($\beta2 = 0.412$; see Model 1).

Hypothesis 2 proposed that a significant relationship existed between inter-personal relationships and knowledge sharing in construction projects, and the results suggested that inter-personal relationships were positively associated with more knowledge sharing ($\beta4 = 0.370$; see Model 2). Hypothesis 3 suggested that knowledge sharing had a positive influence on innovative behaviour in construction projects, and the results showed that knowledge sharing was positively associated with more innovative behaviour ($\beta6 = 0.473$; see Model 3).

Hypothesis 4 proposed that knowledge sharing acted as a mediator between inter-personal relationships and innovative behaviour. Considering the results of Hypotheses 1–3, the results of Hypothesis 4 ($\beta8 = 0.156$; $\beta9 = 0.547$; see Model 4) indicated that knowledge sharing had a partial mediation effect on inter-personal relationships and innovative behaviour.

### 4.3.3. Cross-Level Moderating Effects of Inter-Organizational Relationships

Hypothesis 5 postulated that inter-organizational commitment could augment the influence of inter-personal relationships on innovative behaviour, and the results revealed there was a significant interaction between inter-personal relationships and inter-organizational relationships, which was positively associated with more innovative behaviour ($\gamma11 = 0.353$; see Model 6). Hypothesis 6 proposed that inter-organizational trust could amplify the influence of knowledge sharing on innovative behaviour, and the results revealed there was a significant interaction between knowledge sharing and inter-organizational relationships, and it was positively associated with more innovative behaviour ($\gamma21 = 0.341$; see Model 7).

In addition, as suggested by Andrew Hayes [87], we plotted an interactive graph (Figure 4) to further verify the interaction via estimating the inter-organizational relationship at low level (mean − 1 SD) and high level (mean + 1 SD). Figure 4 consists of two interactive graphs with the slopes for inter-organizational relationship at one standard deviation (SD) below the mean and at one standard deviation (SD) above the mean. As shown by the solid line in Figure 4a, the results suggested that the cross-level moderating effect of inter-organizational commitment relationship is positive and

noticeable, thereby supporting Hypothesis 5. The solid line in Figure 4b shows that the results offer support for accepting Hypothesis 6.

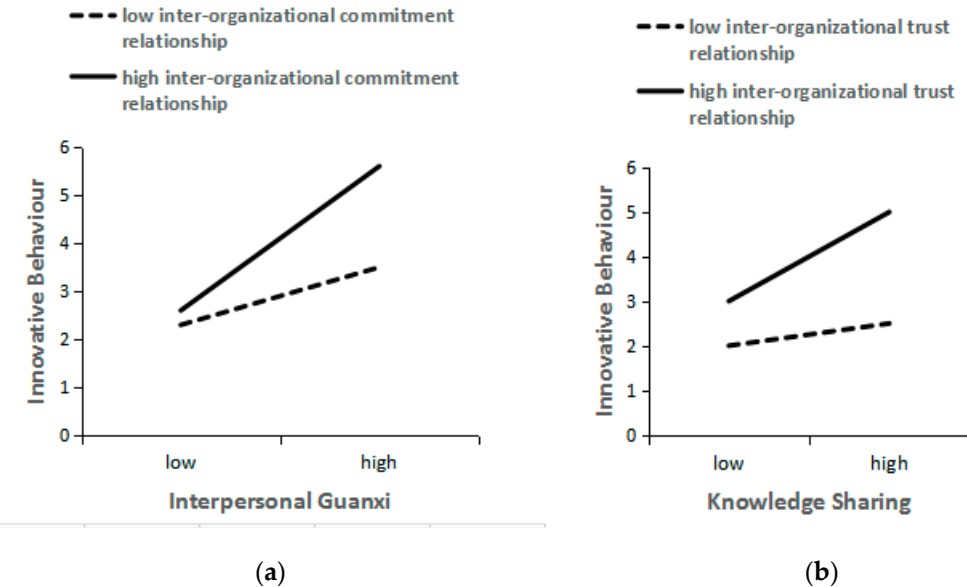

**Figure 4.** The cross-level interaction influence of: (**a**) inter-organizational commitment relationship and inter-personal relationships on innovative behaviour in construction projects; and (**b**) inter-organizational trust relationship and knowledge sharing on innovative behaviour in construction projects.

## 5. Discussion and Implications

### 5.1. Discussion

The increasing importance of inter-personal relationships in innovation management have inspired the researchers to explore the complex mechanism of how inter-personal relationships influence construction innovation. Nevertheless, scholars of innovation in other sectors have also drawn attention to the antecedents of innovation such as guanxi, knowledge sharing and inter-organizational relationship respectively, so there is a need to integrate these antecedents to investigate the mediating and moderating effects on an individual's innovative behaviour. The current study thus investigated how to stimulate innovative behaviour in construction projects through inter-personal relationships, knowledge sharing and the cross-level moderating role of inter-organizational relationships. Based on the research data and analysis, some of the findings of this study are presented below.

Firstly, we found that inter-personal relationships had significant positive influences on innovative behaviour; this is consistent with social capital theory which highlights that guanxi, as a type of social capital can effectively stimulate innovative behaviour [51], and that these influences were more significant in more innovative construction projects. In addition, this research provided evidence that knowledge acts as a partial mediator between inter-personal relationships and innovative behaviour at the individual level. In other words, inter-personal relationships can not only have direct influence on innovative behaviour, but can also have indirect influence on innovative behaviour by knowledge sharing, which is in accord with most previous research [44–46]. The innovative behaviour outcomes in construction projects, originating from in our country-specific sample, show that the inter-personal relationships model can be implemented in the Chinese context.

Secondly, after testifying that inter-personal relationships and knowledge sharing were associated with innovative behaviour, we found there were some individual differences in these antecedents of innovative behaviour, which originated from diverse organizations, as shown by the R square of

Model 4 being higher than that of Model 1 in Table 4. Because different organizations have unique innovation atmospheres, models and policies established in their previous innovative activities, this uniqueness determines the diverse influences of inter-personal relationships and knowledge sharing on individual innovative behaviour.

Finally, building upon those differentiae of influences on innovative behaviour, we further posited that inter-organizational relationships serve as a cross-level moderator, and we utilized Hierarchical Linear Modelling (HLM) to examine cross-level moderating effects. The cross-level results from a heterogeneous sample of individuals in diverse organizations lent support for the role of inter-organizational commitment and trust as inter-organizational relationships associated with innovative and knowledge sharing behaviours. In line with our hypotheses, the influence of inter-personal relationships and knowledge sharing on innovative behaviour varied significantly across organizations, that is, the presence of inter-organizational relationships serves as a cross-level moderator. Actually, the results revealed that inter-organizational relationships in construction projects can amplify the influence of inter-personal relationships on innovative behaviour.

*5.2. Implications*

This research establishes a double-level model to understand individual innovative behaviour in construction projects. In contrast to findings in the extant literature, our double-level conceptual model is integrated by the concepts of inter-personal relationships, knowledge sharing, organizational relationship and individuals' innovative behaviours, which is both fruitful and necessary to understanding innovative behaviour in China-specific construction project settings.

This study is also the first cross-level empirical test of inter-organizational relationships moderating the direct and indirect influence of inter-personal relationships on individuals' innovation behaviour. Prior studies on links between guanxi and innovation have focused on performances at the firm's level [23], but this research tried to bridge the gap by utilizing multi-level analyses to simultaneously consider individual-level and organizational-level variables.

Finally, this research differentiates itself from prior studies because social capital and exchange theories were applied to examine the links between the inter-personal relationships and individuals' innovative behaviour.

Besides the theoretical implications, this research provides crucial guidelines for managing construction innovation activities in China. Firstly, this study has confirmed empirically that knowledge sharing has a mediating role on the effect of inter-personal relationships on innovative behaviour. Thus, encouraging knowledge sharing between members in construction projects is crucial for construction innovation because the total integrated knowledge exceeds each individual's knowledge [88]. This leads to new knowledge for innovation. Consequently, members in construction projects should be ready to open their minds and share their technology, experience and knowledge with their peers in the process of construction innovation. Such a commitment to openness will help to establish a knowledge management system that facilitates individuals' innovative behaviour. Secondly, the cross-level moderating role of inter-organizational relationships on inter-personal relationships and innovative behaviour or knowledge sharing and innovative behaviour shows that inter-organizational relationships could influence inter-personal relationships and knowledge sharing in construction innovation. Inter-organizational relationships are a main contributor to encouraging members to cultivate better inter-personal relationships and to share more knowledge for innovation. Therefore, firms in the Chinese construction industry must provide the conditions that establish and strengthen inter-organizational trust and commitment amongst the project organizations.

## 6. Conclusions, Limitations and Future Research

### 6.1. Conclusions

The integration of guanxi, knowledge sharing and innovation research is fundamental to achieving the key objectives of this study, which were to investigate the influence of guanxi on innovative behaviour in China's construction industry, the partial mediating influence of knowledge sharing, and the cross-level moderating effect of inter-organizational relationships. Firstly, the conceptual model and research hypotheses were developed through a review of the literature and correlative theories. These hypotheses were confirmed by Hierarchical Linear Modelling. The research results demonstrated that inter-personal relationships not only have directly significant effects on innovative behaviour in construction projects, but also have indirectly stimulated effects on innovative behaviour via knowledge sharing. Therefore, knowledge sharing serves as the partial mediator. In addition, inter-organizational relationships augment and influence inter-personal relationships, knowledge sharing and innovative behaviour by cross-level interaction. Our research findings provide useful insights into understanding the importance of inter-personal and inter-organizational guanxi in China for construction innovation.

### 6.2. Limitations and Future Research

Although this study achieved the research aims, it had several limitations that need to be addressed in future research. Firstly, the inter-organizational relationship variable in this study was based on a survey sample of project managers. Although these project managers might have better understanding of external relationships relevant to their organizations, having more members in a variety of roles within each organization in the examination of this variable would enhance the reliability of the survey results. Secondly, the data from the questionnaire surveys for measuring all the variables were obtained simultaneously, rendering it difficult to depict the causal links amongst the variables. Consequently, future research should pay close attention to acquiring longitudinal data to explore the dynamic links amongst guanxi and innovation performance in construction projects. Finally, the interpretation of results in the current study came from only 35 organizations in China. Future research could be carried out with more samples from more organizations in China.

**Author Contributions:** Y.Q., designed this study and completed the paper in English; M.W. provided good research advice and supervision; Y.Z. provided good research advice and writing—review & editing; R.J. writing—review & editing; R.Y. and Q.W. participated in proofreading the paper.

**Acknowledgments:** This study is supported by overseas Study Project in Central South University (2018).

**Conflicts of Interest:** The authors declare no conflict of interest.

## Appendix A

(1) Could you describe what boosted organizational and your initiative to participate in construction innovation at current stage?
(2) How is your organizational innovation culture? Whether is it willing to carry out collaborative innovation with others?
(3) In your opinion, what factors are most critical while facing construction innovation?
(4) What strategies has your organization utilized to acquire resources (e.g., knowledge) and information for innovation?
(5) Did you or your organizations utilize any interpersonal or inter-organizational effect strategies (e.g., inter-personal relationships or inter-organizational relationship)?
(6) How did those strategies benefit your organization in the long term?
(7) How would those strategies you utilized help with motivation or improvement of innovative behavior?

# Appendix B

**Table A1.** Items for each construct of the four latent variables.

| Latent Variables | Measurement Items |
| --- | --- |
| Inter-personal relationships | My organizational main technicians in construction project have good personal relationships with . . . in the process of innovation, or there is at least a kind of guanxi (such as families or friends or classmates or colleagues)<br>a . . . the main technicians of owner<br>b . . . the main technicians of designer<br>c . . . the main technicians of contractor<br>d . . . the main technicians of supplier<br>e . . . relevant key government officials |
| Knowledge sharing | a. The ordinary member of project-based organization is capable of sharing their expertise to bring new initiatives to fruition.<br>b. I feel that I have learned from each other by sharing information or ideas.<br>c. I am willing to share information or ideas with the other member of project-based organization.<br>d. In the project, i am willing to exchange and combine ideas to find solutions to problems. |
| Inter-organizational relationship | Trust from senior managers, being able to represent the organizations:<br>a. We believe the information that this partner provides us.<br>b. We trust this partner keeps our best interests in mind<br>c. This partner keeps promises it makes to our firm<br>d. This partner is trustworthy<br>e. We find it necessary to be cautious with this partner<br>Commitment from senior managers, being able to represent the organizations. We expect relationship to continue for a long time<br>g. We are committed to this partner.<br>h. We expect relationship to strengthen over time.<br>i. Considerable effort and investment in innovation activity. |
| Innovative behaviour | a. The members always generate creative ideas or new solutions<br>b. The members would encourage and champion ideas to others.<br>c. The members explore and secure funds or resources required for implementing new ideas.<br>d. The members establish adequate plans and schedules for implementing new ideas.<br>e. The members would contribute suggestions or approaches for others' creative ideas. |

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
