# Peer review of "Understanding the Double-Level Influence of Guanxi on Construction Innovation in China: The Mediating Role of Interpersonal Knowledge Sharing and the Cross-Level Moderating Role of Inter-Organizational Relationships"

_sustainability, doi:10.3390/su11061657_

Round 1

Reviewer 1 Report

The revised manuscript has been significantly improved.

This paper can now be accepted for publication.

The resolution of figure 4 is low.

Author Response

Comments
   The revised manuscript has been significantly improved, and This paper can now be accepted for publication.
Author’s Responses 
   Thanks for the constructive comments, we are grateful for supportive comments that have been addressed in the revised manuscript.

Comments
   The resolution of figure 4 is low.
Author’s Responses 
   It is really true as Reviewer 1 suggested that the resolution of figure 4 is low. We have replaced the original figure 4 with higher resolution in the paper.

Reviewer 2 Report

This research used social capital theory and social exchange theory to examine Guanxi’s role in motivating innovative behavior in a China-specific construction context. The results showed that inter-organizational relationships augment inter-personal relationships and knowledge sharing on innovative behavior by cross-level interaction. This study enhanced an understanding of Guanxi and innovative behavior in China-specific construction project setting. The reviewer believes that the current version of the manuscript is not yet ready for publication; the authors are encouraged to consider the following comments and suggestion and revise the manuscript accordingly.

1. The authors should consider beefing up the Relevant Work section to include more background for the research. Also, the authors should consider moving the Research Hypotheses section to the Introduction section or the Methodology section. The Introduction section should focus on introducing the research objectives and the research questions that need to be addressed, while the Relevant Work section should focus on literature review of related work and defining the research gap.

2. The reviewer suggests the authors consider adding a section that focuses on discussing the future application of this research. To the reviewer, this research has a China-specific context, but how can we use the research benefit the entire construction industry across the world.

3. How was the questionnaire developed? How to make sure the questionnaire was appropriate for this research? This should be discussed in the manuscript.

4. What is the rationale behind the selection of the various organizations to be interviewed? Why were these criteria selected? Are there any literature supporting this selection? How did the authors know these criteria were sufficient?

5. Some of the figures need to be revised to make them more legible. For example, Figure 4 needs to be redone. It is very blurry and hard to read.

Author Response

Comments

1. The authors should consider beefing up the Relevant Work section to include more background for the research. Also, the authors should consider moving the Research Hypotheses section to the Introduction section or the Methodology section. The Introduction section should focus on introducing the research objectives and the research questions that need to be addressed, while the Relevant Work section should focus on literature review of related work and defining the research gap.

Author’s Response

   Thanks for the reviewer 2’s comments. 

   As for introduction section, we had proposed the main objective of this study and have added the research questions in the third paragraph [page 2], for instance, The main objective of this study is to acquire an intensive understanding of the influence of guanxi on individual innovative behaviour in construction projects, and to reveal the nature of the mediating role of knowledge sharing and cross-level moderation role in inter-organizational collaborative relationships. The research questions are how guanxi influences construction innovation at the double levels and what is the role of knowledge sharing in construction innovation. 

   As to Relevant Work section, we agree with reviewer 3‘s opinion that literature review of related work and defining the research gap should be focused on. However, we had introduced literature review and found the gaps in introduction section and theoretical background.

   Meanwhile, the reviewer 2 suggest that we consider moving the Research Hypotheses section to the Introduction section or the Methodology section. We think that moving it to the Methodology section is fit, but if we move it, the Methodology section will be too long to affect the layout of the chapter and the aesthetics of the structure of the article.

   In summary, according to similar high quality journal articles(Guanxi, IT systems, and innovation capability: The moderating role of proactiveness), we decided to change the title of Relevant Work section into Theoretical background, Research Hypotheses and Conceptual Model Development,which can help make the structure of the article more reasonable 

2. The reviewer suggests the authors consider adding a section that focuses on discussing the future application of this research. To the reviewer, this research has a China-specific context, but how can we use the research benefit the entire construction industry across the world.

Author’s Response

   This is a good suggestion, the question:”how can we use the research benefit the entire construction industry across the world” deserves deep thinking. The scope of this issue is relatively wide, and adding a section is hard to clearly explain the research benefit of the entire construction industry across the world. Therefore, we will focus on this issue to write a article in the future. And the discussion of the research benefit in this paper is involved in management guide section of 5.2. For instance, encouraging knowledge sharing between members in construction project is crucial for construction innovation. And Firms in the Chinese construction industry must provide the conditions that establish, strengthen inter-organization trust and commitment amongst the project organizations .

3. How was the questionnaire developed? How to make sure the questionnaire was appropriate for this research? This should be discussed in the manuscript.

Author’s Response

   As for comment 3, we had made a relevant explanation in Design of Questionnaire section and add some discussions in this section. For example, The questionnaire, developed from the literature review and initial interviews, was separated into two parts. The first part consisted of respondents’ personal information (i.e. the gender, education level and working life) and measured the respondent’s innovative behaviour (I.I.N., containing five items). The second part measured three antecedents of innovative behaviour, including inter-personal relationships (I.R., containing five items), knowledge sharing (K.S., containing four items) and inter-organizational relationships (I.O.R, containing nine items). These contents can make sure the questionnaire was appropriate for this research. 

4. What is the rationale behind the selection of the various organizations to be interviewed? Why were these criteria selected? Are there any literature supporting this selection? How did the authors know these criteria were sufficient?

Author’s Response

   As for comment 4, we had made a relevant explanation in Sample Distribution section. For the question:”What is the rationale behind the selection of the various organizations to be interviewed? “, the literature[76] pointed out that it is a project-based industry, construction involves many participants, including government agencies, owners, designers, construction units, suppliers of materials and equipment, universities and scientific research institutions, and each has diverse roles in the process of construction innovation.

   As to the question:”Why were these criteria selected? ”, these participants had to fulfill the following conditions: (1) belonged to a basic functional unit in construction innovation, (2) had considerable experience of construction innovation or innovation management, and (3) frequently worked with some of the other participants. After many research seminars with Chinese experts on construction innovation, the selection of these conditions is derived from their understanding and suggestions on construction innovation.

   As to the questions:”Are there any literature supporting this selection? How did the authors know these criteria were sufficient?”, there is few literature supporting this selection, and After many research seminars with Chinese experts on construction innovation, the selection of these conditions is derived from their understanding and suggestions on construction innovation.

5. Some of the figures need to be revised to make them more legible. For example, Figure 4 needs to be redone. It is very blurry and hard to read.

Author’s Response

Thanks for the reviewer 2’s suggestion, and we have revised the Figure 4 in the paper

Reviewer 3 Report

Thanks for inviting me to review this paper. This paper reports a research on the influence of relationship on construction innovation in China through analysis of survey results. 

I did not see sufficient merits of this work. There are already numerous relevant studies, and I could not clearly identify what new knowledge this research brings. The analysis of the results is not well interpreted. Besides, the logic and organization of this research are unclear. The quality of presentation is low. The authors even did not clear the track of revision. The figures and tables also look not professional. 

Author Response

Thanks for the reviewer 3’s suggestion.
  As for merits of the research and what new knowledge this research brings, we have discussed theoretical implications and management guide in 5.2 implication section, and from the introduce section, although there are already numerous relevant studies, we found that the research on construction innovation at double levels remains in its infancy. The analysis of the results is interpreted according to HLM calculation, and the logic and organization of this research referenced similar high quality journal articles(Guanxi, IT systems, and innovation capability: The moderating role of proactiveness) and have slightly revised in the paper, and we have used the "Track Changes" function in Microsoft Word to make sure that changes are easily visible to the editors and reviewers.The figures and tables would be set according to figures and tables other similar high quality journal articles(Guanxi, IT systems, and innovation capability: The moderating role of proactiveness).

Round 2

Reviewer 2 Report

The authors have address all my comments and questions. 

Author Response

  Thanks for your comments.

Reviewer 3 Report

The quality is improved, but the quality of presentation is still low. Further revision must be made. Here are some suggestions for improvement:

(1) Please modify Table 3. I do not know how the description corresponds to the models. Also, I do not know how to read Table 4.

(2) Inconsistent citation formats are used. See: lines 204, 208, 353, 361, 372, 374, etc.

(3) Professional presentation should be used. In lines 433 and 434, how did you define the symbols? What do you mean by "p<.001" in line 434? From line 437 to 454, what do you mean by the symbol *?

Author Response

(1) Please modify Table 3. I do not know how the description corresponds to the models. Also, I do not know how to read Table 4.

Authors Responses

   Thanks for your suggestions. The table 3 have been modified, the models involves diverse factors, for instance, model just incaluded two factors, and it can be descripted through innovative behaviour with knowledge sharing= β+ β× guanxi (integrated: interpersonal relationships and organizational relationships ) .

   As for table 4, in fact, by analyzing the formula in the above paragraph of Table 4 and the explanation for table 4 in the next paragraph, it is better to understand the contents of Table 4. For instance, H4: The mediation of inter-personal relationships and innovative behaviour by knowledge sharing, it involves formula that innovative behaviour = β7(3.613) + β8(0.156)× inter-personal relationships + β9(0.547)× knowledge sharing + ε3 , and these coefficients correspond to the data in Table 4.

(2) Inconsistent citation formats are used. See: lines 204, 208, 353, 361, 372, 374, etc.

Authors Response

   Many thanks, and we have revised them in the paper.

(3) Professional presentation should be used. In lines 433 and 434, how did you define the symbols? What do you mean by "p<.001" in line 434? From line 437 to 454, what do you mean by the symbol *?

Authors Response

  There is no "p<.001" in line 434. And we find "p<.001" in line 434 and 434. It represents It represents that the measurement results are significant.

  From line 437 to 454, what do you mean by the symbol *? , as fo the symbol *,it is a multiplication symbol. In fact, it is not appropriate to use symbol *to represent multiplication, and we have replaced symbol * with symbol×.

This manuscript is a resubmission of an earlier submission. The following is a list of the peer review reports and author responses from that submission.

Round 1

Reviewer 1 Report

The article is sufficiently novel and interesting to warrant publication and it adheres to the journal's standards. The article is clearly laid out. All the key elements are present: abstract, introduction, methodology, results, discussion and conclusions. The title clearly describes the article and the abstract reflects the content of the article. The introduction contains a brief description of the actual state-of-the-art, and clearly state the problem being investigated. The authors accurately explain what they discovered in the research. The claims in conclusion are supported by the results. The references are accurate.

Reviewer 2 Report

The article entitled “Understanding the Influence of Interpersonal Guanxi on Construction Innovation in China” examines guanxi’s role in motivating individuals’ innovative behaviour in China-specific construction context and verifies the significance of guanxi at the process of stimulating innovative behaviour. It hypothesizes the main effects of interpersonal guanxi on innovative behaviour, the mediating effect of knowledge sharing, and the cross-level moderating effect of inter-organizational relationship. Then the hypotheses are tested by a survey which receives 178 responses from 35 different organizations. The results are analyses using Hierarchical Linear Modelling (HLM) and reveal that interpersonal guanxi has positive influences on innovative behaviour, highlighting the partial mediating effects of knowledge sharing.

Some editing of the English language and style is required.

The resolution of most figures is low.

The subject is of limited interest to the readers.

Reviewer 3 Report

This paper aims how guanxi could affect individuals’ innovative behaviour. However, the paper is not well organized and the contributions are not clear. There are many grammar mistakes which hinder readers' understanding. I am also confused with the research methodology. My concerns are listed as follows:

The authors failed to explain the difference between the guanxi model in Chinese construction industry and the social capital theory and social exchange theory in general. Otherwise, why the previous model cannot be applied here? What are the features that make your guanxi model suitable for the analysis for individuals’ innovative behavior in Chinese construction industry?

The authors' model contains multiple dimensions. But the paper doesn’t analyze the relationships among the dimensions.

On Page 1, there are many grammar mistakes. The first paragraph seems to be irrelevant to the paper.

On Page 2, in the 3rd paragraph, there are multiple objectives. The paper claims to “introduce guanxi into a construction innovation management”. The entire paper fails to provide any discussion on how to do this. The last paragraph in “Introduction” can be removed.

On Page 3, the literature review is very unorganized. The authors admitted that “previous studies have discussed the link…” What is the link? How is the relationship built in this paper different from the previous ones?

On Pages 4-5, all the hypotheses seem to be common senses. It is hard to see the contribution of this paper to the body of knowledge.

Many sentences have grammar mistakes. I suggest that the author should find an English tutor to thoroughly revise the English in this paper before submission.

Considering the contribution, model formulation as well as the English expression, I think this paper is not suitable for publication.